# Current Methods for the Detection of Antibodies of Varicella-Zoster Virus: A Review

**DOI:** 10.3390/microorganisms11020519

**Published:** 2023-02-17

**Authors:** Dequan Pan, Wei Wang, Tong Cheng

**Affiliations:** State Key Laboratory of Molecular Vaccinology and Molecular Diagnostics, National Institute of Diagnostics and Vaccine Development in Infectious Diseases, School of Life Sciences, School of Public Health, Xiamen University, Xiamen 361102, China

**Keywords:** varicella-zoster virus (VZV), antibody detection methods, FAMA, ELISA

## Abstract

Infection with the varicella-zoster virus (VZV) causes chickenpox and shingles, which lead to significant morbidity and mortality globally. The detection of serum VZV-specific antibodies is important for the clinical diagnosis and sero-epidemiological research of VZV infection, and for assessing the effect of VZV vaccine immunization. Over recent decades, a variety of methods for VZV antibody detection have been developed. This review summarizes and compares the current methods for detecting VZV antibodies, and discussed future directions for this field.

## 1. Introduction

Varicella-zoster virus (VZV) is a highly contagious alpha-herpesvirus that infects more than 90% of people worldwide [1,2]. Chickenpox (varicella) is the outcome of primary infection with VZV and mainly affects children. As per the report published by the World Health Organization (WHO) in 2014, the minimum of the annual global disease burden of chickenpox was estimated to be 140 million cases, of which 4.2 million have severe complications leading to hospitalization and death [3]. Although usually a mild and self-limiting illness, chickenpox poses a greater risk of severe disease to pregnant women, neonates, VZV-seronegative adults, and immunocompromised individuals [4]. The reactivation of latent VZV causes shingles (herpes zoster), which occurs mainly in people ≥50 years of age and is usually associated with intense neuralgia [5]. It was estimated that about one third of individuals who have had chickenpox will develop shingles [6]. The incidence and severity of shingles increases with age or immunosuppression [7]. Recent studies have shown that COVID-19 vaccination may also increase the risk of VZV reactivation and thus potentially increase the incidence of shingles, especially in the elderly population [8,9,10]. To date, there is still no specific cure for VZV-induced diseases. Vaccination is among the most cost-effective ways for preventing chickenpox and shingles. The most widely used chickenpox vaccines consist of the Oka strain of live-attenuated VZV (vOka), and there are currently two kinds of shingles vaccines on the market, including a live-attenuated vaccine also based on vOka (ZOSTAVAX^®^, Merck Sharp & Dohme, Rahway, NJ, USA) and a recombinant subunit vaccine based on VZV glycoprotein E (Shingrix^®^, GlaxoSmithKline, Brentford, Middlesex, UK) [6,11,12]. However, the use of these vaccines is not universal and thus a large proportion of the global population has not been vaccinated against VZV. Consequently, VZV infection is still prevalent and accounts for a significant disease burden worldwide.

Laboratory testing is essential for the diagnosis and surveillance of VZV-induced diseases. Virus isolation was once the “gold standard” for the diagnosis of VZV infection [13,14]. However, this method is time-consuming and not readily accessible, and has thus been replaced by polymerase chain reaction (PCR) for the detection of viral DNA and direct fluorescence assays (DFA) for the detection of viral proteins [15,16]. Most recently, a recombinase-aided amplification-lateral flow system (RAA-LF) has been used for the rapid detection of VZV DNA, which is easier to use and requires no equipment [17]. PCR is currently regarded as the most sensitive and reliable method for VZV detection [14]. Meanwhile, serological assays, which detect VZV-specific antibodies, also represent reliable diagnostic tools in the detection of VZV infection, and have been implemented as complementary approaches to PCR. Furthermore, since the quantitative detection of anti-VZV antibody is necessary for measuring the infection history and evaluating the immune status against VZV in the population, serological assays have been widely used in epidemiological investigations of VZV and evaluations of immune responses to VZV vaccination. These assays include the fluorescent-antibody-to-membrane-antigen (FAMA) test, the complement fixation (CF) test, the immune adherence hemagglutination assay (IAHA), the latex agglutination (LA) test, the radioimmunoassay (RIA), the neutralization assay, the chemiluminescent immunoassay (CLIA), the enzyme-linked immunosorbent assay (ELISA), the immunofluorescence assay (IFA), the time-resolved fluorescence immunoassay (TRFIA), and the lateral flow immunochromatographic assay (LFIA). Among these tests, FAMA and ELISA are the most widely used. On the other hand, the CF test has low sensitivity, while RIA uses harmful radioactive materials; therefore, neither method has been widely used in recent years.

Herein, we review the existing serological assays for the detection of anti-VZV antibodies and compare their advantages and limitations. This paper could help clinicians and technicians to choose the appropriate serological method for diagnosing VZV infection or assessing the efficacy of VZV vaccines.

## 2. Methods for the Detection of Anti-VZV Antibodies

### 2.1. FAMA

The FAMA test, which was initially developed by Williams et al. [18], is the most extensively validated assay and is considered the “gold standard” for VZV antibody detection [19,20]. This method determines the presence of antibodies specific to viral proteins that distribute on the surface of VZV-infected cells, which correlate with protection from disease. Different human cells, including HFF [21], MRC-5 [22], Vero [23] and Raji [24], have been used to perform the FAMA test in previous studies. According to the standard FAMA procedure (schematic shown in Figure 1A), cells in culture are infected with VZV for 48–72 h and harvested by trypsin digestion until 70–90% of them show a cytopathic effect. Then, the infected cells are resuspended in PBS and incubated with serial dilutions of human sera to bind with VZV-specific antibodies. Following washing, cells are incubated with a fluorescein-conjugated anti-human secondary antibody (specific for IgG, IgM, or IgM and IgG). After a second washing step, the cells are transferred into small wells on glass slides and incubated for certain time to allow the cells to attach. Finally, cells are sealed with 90% glycerin and a cover glass before observation under a fluorescence microscope [18]. When the viral proteins (e.g., VZV glycoproteins) distributed on the surface of VZV-infected cells bind to their specific antibodies in the serum samples, ring-like fluorescence patterns are typically observed. The highest dilution that can still cause a positive ring-like fluorescence reaction is taken as the FAMA titer of the serum sample. Given the false-positive results caused by the non-specific reactivity of undiluted sera, serum samples with a titer of ≥1:2 or 1:4 are generally considered positive [18,25,26]. Several studies have shown that healthy children can be protected from chickenpox infection when they have serum FAMA titers of ≥1:4 for VZV [27,28].

The FAMA test can be performed using chemically fixed VZV-infected cells. The fixed-cell FAMA was developed from the classic FAMA to increase throughput and efficiency and has been used to study human immunity to VZV [29,30,31,32]. In this modified FAMA (schematic shown in Figure 1B), VZV-infected cells are first fixed on the slide with cold acetone [31,32] or glutaraldehyde [29,30], while the other steps and the cutoff values (1:2 or 1:4) are similar to or the same as those in the classic live-cell FAMA test. The acetone fixation increases membrane permeability and allows antibody access to the cell cytoplasm and binding to other viral proteins besides membrane antigens [33]. In comparison, glutaraldehyde fixation does not change the permeability of the cell membrane and detects only the membrane antigen, but changes the natural conformation of the antigen, resulting in reduced sensitivity to a certain extent [19]. The fixed-cell FAMA has several advantages over the classic FAMA, e.g., the slides with attached cells can be prepared in large batches in advance and stored in a freezer for a long time, which enables testing at any time and reduces hands-on time. In addition, the used FAMA slides can be kept for re-reading, which creates the possibility of them being inspected by the drug administration. However, the specificity of fixed-cell FAMA could be challenged, since the procedure of fixed-cell FAMA is similar to that of indirect fluorescent antibody test (IFAT), which also uses fixed VZV-infected cells, and previous studies have documented that serum samples from children weakly cross-reacted between VZV and herpes simplex virus (HSV) in IFAT [19,34,35]. Nonetheless, the influence of cell fixation on the specificity of FAMA remains controversial and needs to be further clarified.

To avoid the use of infectious viruses, one study has reported a simple and safer FAMA using HEK293T cells transfected with a plasmid encoding VZV glycoprotein E (gE) to replace VZV-infected cells, which is called the gE FAMA (schematic shown in Figure 1C) [36]. The gE FAMA exhibited a similar staining effect to classic FAMA, and the gE-FAMA titers were closely correlated with the gp-ELISA data. However, this assay only detects antibodies against VZV gE, and the abundance of expressed gE on plasmid-transfected cells is different from that of VZV-infected cells; the cutoff value of gE FAMA may be different from that of classic FAMA, and remains to be determined.

All the above-mentioned FAMA assays rely on experienced technicians making result judgments under fluorescence microscopy, which not only leads to subjective bias but also limits the throughput of the detection. Some studies have reported a flow cytometry-adapted FAMA (flow FAMA; schematic shown in Figure 1D), in which flow cytometry is used instead of examination under a microscope to analyze the fluorescence-labeled cells [37,38,39]. The positive cutoff value determined by flow cytometry analysis can make the judgment of FAMA more objective, and the automated measurement can also reduce the complexity of operations and increase the detection throughput of FAMA. In a study involving 62 human serum samples, the detection accuracy of flow FAMA was 90.32% compared with that of standard FAMA [38]. In the flow FAMA, the matter of how to select the appropriate cutoff value remains a key problem. In addition, a flow cytometer is required for the flow FAMA, which limits the application of this method.

Taken together, the classic live-cell FAMA test is regarded as the “gold standard” to detect anti-VZV antibodies because of its high sensitivity and specificity. However, the standard FAMA procedure is semi-quantitative, low-throughput, and labor-intensive, and requires a subjective evaluation by trained, experienced technicians, which limits its widespread use. Fixed-cell FAMA and gE FAMA are modified from the classic live-cell FAMA to improve the throughput and safety, but both of them have their own defects, including possibly lower specificity, a lack of validation, or ambiguous cutoff values (shown in Table 1).

### 2.2. ELISA

ELISA is one of the most common antibody detection methods and has been widely used for the quantitative detection of anti-VZV antibodies for epidemiological investigations of VZV infection and for efficacy evaluation of the varicella and zoster vaccines [25,40,41]. Indirect ELISA is the most commonly used type. According to the procedure of indirect ELISA (schematic shown in Figure 2A), VZV antigens are coated on 96-well polystyrene ELISA plates and subsequently blocked with bovine serum albumin or normal goat serum. Diluted serum samples are then added into the plate wells. After incubation and washing, peroxidase- or alkaline-phosphatase-conjugated anti-human antibodies are added to detect the captured anti-VZV antibodies. After another washing step, the substrate solution is added for a chromogenic reaction, which is then terminated with excess acid or base. Finally, the optical density (OD) or absorbance value of appropriate wavelength is measured quantitatively using a spectrophotometer. Within a certain range, the OD or absorbance value is proportional to the number of binding antibodies on the plate, so the standard curve can be drawn through the detection of standard products and achieve quantitative detection of anti-VZV antibodies [42,43,44].

Nowadays, there are several commercial VZV antibody ELISA kits available [19,45]. They are mostly indirect ELISAs and use either whole VZV-infected cell lysate (WC ELISA) [25,44] or purified glycoprotein (gp-ELISA) [46,47] as the antigen to capture anti-VZV antibodies. Different ELISA kits are calibrated according to the first international standard for varicella-zoster immunoglobulin, and results of <50 mIU/mL are considered negative, but the cutoff value varies [25,41,48]. The majority of these commercial ELISAs are designed to measure antibody levels after natural infection and are found to be insufficiently sensitive to measure antibody responses to chickenpox vaccination [19,49,50]. One study compared four commercial ELISAs and showed that their sensitivity ranged from 60.4% to 91.8%, values that are low compared with those of the FAMA test [25]. To address the issue of sensitivity, Merck has developed an in-house, highly sensitive, and specific gp-ELISA that uses lentil lectin-purified VZV glycoproteins, including gE, gB, and gH, from VZV-infected cells as the antigen [51]. The Merck gp-ELISA has been used extensively to evaluate antibody responses in children immunized with the Varivax Oka vaccine and a titer of 5 gp-ELISA units/mL (equivalent to 10 mIU/mL, by the international reference standard) was found to be associated with a high degree of protection against breakthrough infection during seven follow-up years [52,53]. However, the Merck gp-ELISA is not commercially available and is restricted to only a few specialist testing centers. In general, the gp-ELISA has high sensitivity and specificity, and a high consistency with FAMA, and is thus considered to be the most likely alternative to FAMA [54]. In addition, GSK also developed an in-house ELISA to detect VZV gE antibody (gE-ELISA) and applied it to evaluating the immunogenicity of a herpes zoster subunit vaccine Shingrix [55].

In addition to indirect ELISA, some researchers have developed a competitive ELISA and a gE double-antigen sandwich ELISA for VZV antibody detection [56,57]. Firstly, for the competitive ELISA (schematic shown in Figure 2B), an anti-ORF9 antibody is used to capture VZV particles, and then an HRP-labeled anti-gE antibody is used to compete with serum anti-VZV antibodies for virus binding. Using the following formula: PI (%) = 100 × [1 − (positive serum OD450/negative reference serum OD450)], the blocking rate can be calculated to evaluate the VZV antibody levels in serum samples. The study showed that the competitive sandwich ELISA had a sensitivity of 95.6%, a specificity of 99.77%, and coincidence of 97.61% compared to the FAMA test. Secondly, the gE double-antigen sandwich ELISA (schematic shown in Figure 2C) is modified from the gE-based indirect ELISA by replacing the enzyme-conjugated anti-human secondary antibodies with HRP-labeled recombinant gE protein. The study showed that the gE double-antigen sandwich ELISA had sensitivity of 95.08% and specificity of 100% compared to the FAMA test. The findings in these studies suggest that the use of antibody competition or a dual-gE-antigen sandwich could increase the sensitivity and specificity of ELISA for VZV antibody detection. However, to date, there are no reports on the further application and validation of these two methods.

Taken together, to date, many types of ELISAs have been developed for anti-VZV antibody detection (summarized in Table 2). Compared to the FAMA test, these ELISAs are quantitative, easy to use, and high throughput, but are considered to be less sensitive, except for the Merck gp-ELISA, which is regarded as sensitive enough to be an optimal alternative reference assay to FAMA. Recent advances have improved the performance of several ELISAs, achieving sensitivity and specificity similar to FAMA, but these new methods still lack further validation or else are not commercially available.

### 2.3. Neutralization Assay

The neutralization assay measures the titers of neutralizing antibodies that confer protection from VZV infections. However, the sensitivity of the earliest neutralization assay was low, making it difficult to detect anti-VZV neutralizing antibodies in individuals many years after infection [58]. Some studies have reported that the addition of guinea pig complement and anti-immunoglobulin antibodies can make the sensitivity of the neutralization assay for VZV 2 to 16 fold and 7 to 100 fold higher, respectively, and thus the enhanced neutralization assay is more frequently used for the evaluation of antibody responses to VZV infection, compared to the original method [59,60,61,62,63].

According to the procedure of the complement-enhanced neutralization assay for VZV (schematic shown in Figure 3) [59,64,65], which is modified from the universal plaque-reduction neutralization test (PRNT), hundreds of PFUs of cell-free VZV are mixed with diluted heat-inactivated serum samples (at 56 °C for 30 min) and guinea pig complement, and co-incubated at 37 °C for 1 h before being added into the cultured cells (e.g., MRC-5). After incubation for 5–7 days, the number of virus plaques is directly counted under an inverted microscope, and the highest dilutions of serum that result in ≥50% reduction in plaque counts are defined as the neutralization titers. In this experiment, cells can be stained with dyes such as crystal violet to make the plaques easier to observe. The neutralization assay requires viral plaque formation and takes about one week to complete. To reduce the testing time, an indirect immunoperoxidase assay (IPA) was used to stain VZV-infected cells and shortened the test period to 72 h [66,67]. Another study established a neutralization test basing on an enzyme-linked immunosorbent spot (Elispot) assay with VZV-gK protein as the detection target, and shortened the test period to 36 h [68].

Taken together, neutralization assays can directly detect the presence of neutralizing antibodies to various types of viruses in sera, and have been widely used to determine virus infection and evaluate the protective efficacy of vaccines. However, this conventional method has a low sensitivity when detecting anti-VZV neutralizing antibodies, possibly because VZV has a highly cell-associated nature and grows to low titers in culture, thus affecting the interaction between serum antibodies and cell-free VZV particles. Furthermore, the test period of the neutralization assay is relatively long, and the operation is labor-intensive and low-throughput. Given all these limitations, the neutralization assay for VZV is not commonly used nowadays. Nonetheless, neutralization assays in combination with high-sensitivity immunodetection methods (e.g., Elispot) have, in recent years, shown promise in achieving high-throughput quantitative analysis of anti-VZV neutralizing antibodies.

### 2.4. IFA

IFAs for the detection of anti-VZV antibodies include the anti-complement immunofluorescence (ACIF) assay [69] and the IFAT [70]. Since the procedure of IFAT is similar to that of FAMA using fixed cells, and detects not only viral glycoproteins on the surface of infected cells but other VZV antigens within them, they are generally regarded as the same immunoassay. For the ACIF assay, the complement is mixed with diluted serum samples before incubation with chemically fixed VZV-infected cells, and then the bound complement is detected with fluorescence-conjugated anti-C3. Positive and negative samples can be confirmed by comparing the fluorescence of infected and uninfected cells, and, like the FAMA test, the highest dilution that causes a positive fluorescence reaction is regarded as the antibody titer of the serum sample [69]. For the IFAT, the only difference is that fluorescence-labeled anti-human secondary antibodies are directly used for detection without the aid of a complement. Although it has been reported that IFA is more sensitive than FAMA, its specificity seems problematic since a cross-reaction was found with other herpesviruses, such as HSV [71]. The same doubt about specificity also exists in FAMA using fixed cells. Nevertheless, the specificity of these immunoassays can be improved by adjusting the experimental methods and materials. For example, Sauerbrei et al. used fixed VZV-infected A549 cells in IFA for VZV antibody detection (schematic shown in Figure 4), which showed high specificity without cross-reaction with anti-HSV antibodies and was 100% consistent with FAMA [31].

### 2.5. TRFIA

TRFIA, which is also called dissociation enhanced-lanthanide fluorescence immunoassay (DELFIA), was first used to detect anti-VZV antibodies in 2006 [72]. As shown in Figure 5, the procedure of TRFIA is similar to that of ELISA. Firstly, purified VZV antigens are coated on DELFIA microtiter plates, and, after washing, the plates are incubated with diluted sera. Then, the plates are incubated with europium (EU)-labeled anti-human IgG conjugate as a secondary antibody to form EU-labeled antibody-antigen immune complexes, and the DELFIA enhancement solution is added to enhance the fluorescence signal of EU3+. Finally, the fluorescence signal is captured by a DELFIA plate reader, and the concentration of VZV antibodies in the serum samples is further calculated according to the standard curve. The standard curves can be determined with international standard VZV antibodies, so the results of TRFIA can be expressed in the international standard unit (mIU/mL). Some researchers consider 150 mIU/mL to be the suitable cutoff value for TRFIA, since antibody concentrations of >150 mIU/mL seem to provide a protective effect, while others suggest that 130 mIU/mL is sufficient to distinguish between uninfected and infected individuals with TRFIA [73,74].

TRFIA uses EU3+ as the fluorescent probe; it has a long decay time, and antibody concentrations can be measured after other fluorescent substances with short half-lives are decayed, thus eliminating the interference of non-specific fluorescence. Coupled with the narrow emission light crest of lanthanum fluorescence, the background of TRFIA is further reduced. TRFIA has sensitivity and specificity equivalent to those of the Merck gp-ELISA. Furthermore, compared with FAMA, this method is easy to use and has a relatively short testing time. However, TRFIA requires special equipment and is only used in a few specialist testing centers.

### 2.6. IAHA

IAHA was once commonly used to detect anti-VZV antibodies for the evaluation of the immune status of the population against VZV and the immune effect of chickenpox vaccination [54]. IAHA is performed by mixing VZV antigens, serum, a complement, and human type O red blood cells (RBCs), and a positive reaction is indicated by the agglutination of RBCs, which is mediated by their surface C3 receptors [75]. Since the sensitivity of IAHA is low, its application has great limitations.

LA is a modified version of IAHA. According to the procedure of LA (schematic shown in Figure 6), serially diluted serum samples are added to the synthetic latex particles coated with VZV antigen (e.g., VZV gE), and the test samples are determined as positive by the observation of the agglutination reaction [27]. Serum with a titer of ≥1:2 by LA is considered positive [76]. LA is convenient and fast to operate, requires no special equipment, and is commercially available. The sensitivity of LA is almost as good as that of FAMA and is better than that of a commercial ELISA. However, false-positive results may occur in the detection process since LA cannot distinguish between IgG and IgM, and it is difficult to automate and difficult to use on a large scale.

### 2.7. CLIA

CLIA has also been used to detect anti-VZV antibodies. According to the procedure of CLIA (schematic shown in Figure 7), magnetic particles are first coated with purified VZV antigens (e.g., glycoproteins). After incubation with diluted serum samples, secondary anti-human IgA/IgM/IgG antibodies conjugated with isoluminol or acridinium are used to detect the captured VZV-specific antibodies. Next, chemiluminescent detection reagents are added to produce a signal, and the relative light units (RLU) are measured using a full-automatic chemical luminescence immune analyzer and converted to the antibody concentration according to the standard curve [77,78]. The reported CLIA used cutoff values of 150 mIU/mL [79] and 100 mIU/mL [80]. It has been reported that women with CLIA values of <100 mIU/mL are more likely to develop varicella than those with values of >100 mIU/mL. Thus, a value of 100 mIU/mL may distinguish women who are susceptible to chickenpox infection from those who are protected from exposure. However, while emphasizing the importance of this value, some international guidelines note that the CLIA cutoff may vary depending on vaccination status, race, or age.

The CLIA is easy to operate, can achieve automated detection, and has a commercial kit. However, the sensitivity of CLIA still needs to be improved. Recently, a VZV gE-CLIA has shown better sensitivity and specificity than the gp-ELISA (Abcam, Cambridge, MA, USA) [78], but further application and validation of this method are still required.

### 2.8. LFIA

LFIA, a simple point-of-care testing (POCT) based on antigen and antibody immune responses, has been used for the detection of VZV antibodies [81]. The reported VZV LIFA is a paper system that uses truncated VZV gE protein as the capture antigen and consists of a substrate, nitrocellulose membrane, sample pad, binding pad, test line, and control line [82,83]. The protocol for this assay is shown in Figure 8. First, 15 to 20 μL of serum or whole blood is added to the sample port, filtered through the blood separation membrane, and absorbed by the test strip. Then, three drops of chase buffer are added to allow migration of colloidal gold-conjugated goat anti-human IgG and the sample onto the test line. The mixture contacts the test line and the control line in turn, and anti-VZV gE antibodies present in the sample will be captured by the VZV gE protein in the test line, resulting in the visualization of the test line along with the control line after 15 to 20 min [81]. A comparative study between Viro VZV IgG LFIA and the Diamedix VZV IgG ELISA showed that VZV LFIA was more sensitive than ELISA, while having comparable specificity [81]. LFIA provides a simple, inexpensive, and rapid method for VZV antibody detection without equipment and sample pretreatment. However, this detection method cannot achieve the absolute quantitative detection of antibodies.

## 3. Concluding Remarks and Perspectives

In recent decades, a variety of serological methods for detecting anti-VZV antibodies have been established for the clinical auxiliary diagnosis of VZV infection; these methods have facilitated VZV-related epidemiological studies and vaccine studies, as well as risk assessments of healthcare workers. FAMA is the most widely recognized and commonly used method due to its relatively high sensitivity and specificity. However, standard live-cell FAMA is labor-intensive with low throughput, and is susceptible to subjective judgment. ELISA is the most accessible method, but commercial ELISA kits may yield false-positive or false-negative results and are not reliable in evaluating serum conversion after vaccination for chickenpox, which generates lower antibody levels compared to the wild-type VZV infection. Neutralization tests can directly evaluate the immune protection effect, but have a low sensitivity for detecting anti-VZV antibody. IFA and LA are very sensitive but have limitations in determining positive reactions, which make them less than ideal for both chickenpox susceptibility screening and detecting seroconversion to VZV vaccines. TRFIA and CLIA showed good specificity and sensitivity but require further validation. LFIA is an easy-to-use POCT without the requirement for experienced technicians and equipment, but it only provides qualitative or semi-quantitative results.

The existing VZV serological tests mainly detect IgG antibodies, but the analysis of different antibody subtypes is also of value in the diagnosis of VZV infection. For example, it has been reported that the IgG3 subtype was the main subtype in the recovery period of chickenpox, and the IgGl subtype was the main subtype in the recovery period of shingles [84]. Meanwhile, detection of IgGl and IgG2a represents the activation of Th2- and Th1- type immunity, respectively, and would be helpful in elucidating the immune-protective mechanisms of VZV vaccines [85]. Furthermore, the detection of IgM-and IgA-anti-VZV antibodies is also helpful for the diagnosis of VZV infection in immunocompromised individuals [86]. Although FAMA, ELISA, IFA, TRFIA, CLIA, and LFIA can theoretically be applied to analyze different subtypes of anti-VZV antibodies, the relevant research is still somewhat lacking.

While current methods to detect anti-VZV antibodies are numerous and relatively mature, researchers can still strive to make improvements to obtain better detection efficiency and measurement accuracy, as well as greater convenience. Some potential directions for future development are as follows:(1)Biosensor techniques, such as lateral flow assays and electrochemical assays, are evolving rapidly, and have shown promising prospects in making inexpensive, easy-to-use diagnostic tools (e.g., POCT devices) for quicker sensitive and specific detection of anti-VZV antibodies. These biosensor-based assays would not require elaborate instrumentation and/or a laboratory set-up and would therefore be more accessible to researchers, clinicians, and the general public, thus better meeting the needs of large-scale population screening. They could be useful complements to conventional laboratory tests.(2)For national institutions or testing centers involved in disease control and prevention, it is better to combine high-precision detection equipment with different assay strategies to develop standardized methods for the automated measurement of anti-VZV antibodies, to not only achieve high sensitivity, accuracy, and reliability but also to enable high-throughput, objective, and stable testing.(3)It is hoped that the methods developed in the future will be able to detect different subtypes of anti-VZV antibodies with similarly high sensitivity, which would be beneficial for clinical diagnosis, increasing our understanding of the role of antibody responses in the prevention of VZV infection, and may provide insights into ways to improve the effectiveness of VZV vaccines.

## Figures and Tables

**Figure 1 microorganisms-11-00519-f001:**
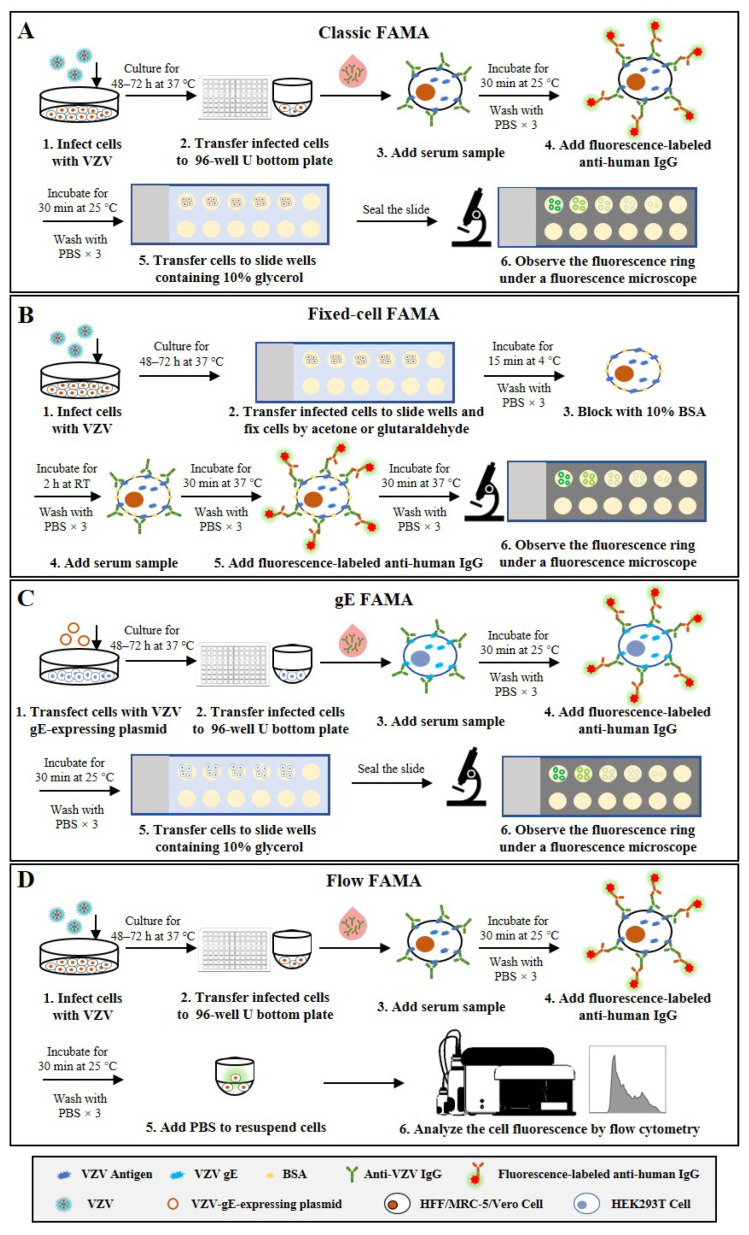
Schematic representation of the basic protocols for four types of FAMA that detect anti-VZV antibodies. (**A**) Classic FAMA using live VZV-infected cells to capture anti-VZV antibodies. (**B**) Fixed-cell FAMA using chemically fixed VZV-infected cells to capture anti-VZV antibodies. (**C**) gE-FAMA using VZV-gE-expressing live cells to capture anti-VZV-gE antibodies. (**D**) Flow FAMA using flow cytometry to analyze the fluorescence-labeled VZV-infected cells.

**Figure 2 microorganisms-11-00519-f002:**
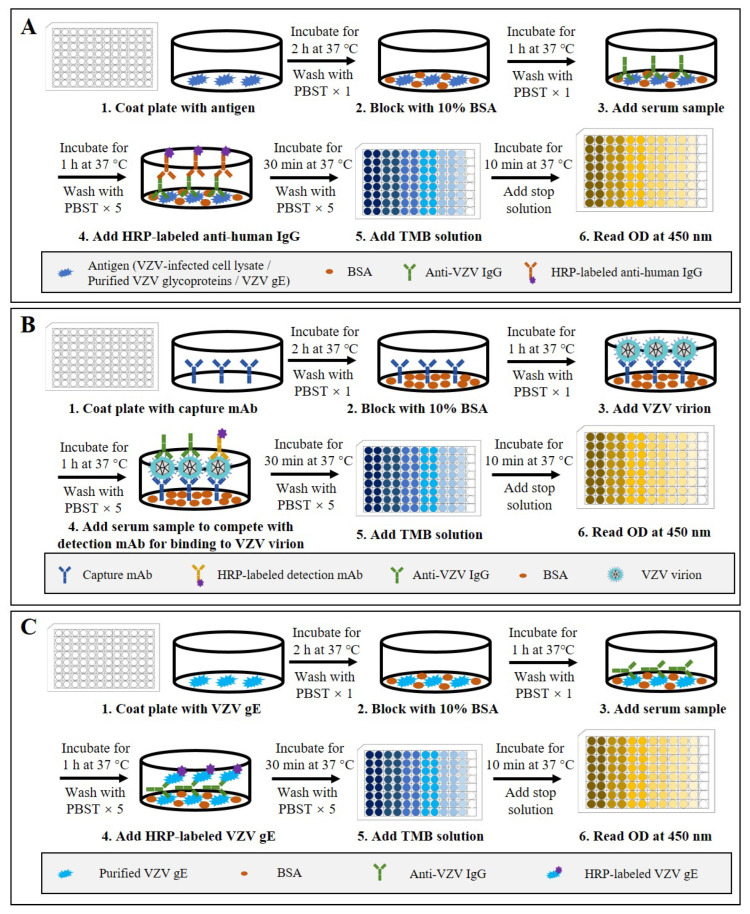
Schematic representation of the basic protocols for three types of ELISA that detect anti-VZV antibodies. (**A**) Indirect ELISA using VZV-infected cell lysate, purified VZV glycoproteins, or VZV gE as the coating antigen. (**B**) Double antibody sandwich competitive ELISA using anti-ORF9 antibody as the detection antibody and HRP-labeled anti-gE antibody as the detection antibody. (**C**) Double gE antigen sandwich ELISA using purified VZV gE as the coating antigen and HRP-labeled gE as the detection antigen.

**Figure 3 microorganisms-11-00519-f003:**
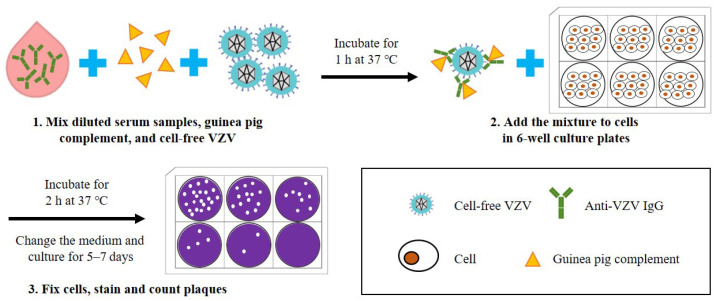
Schematic diagram depicting the basic steps of a neutralization assay for the detection of anti-VZV antibodies. (1) Diluted serum samples, guinea pig complement, and cell-free VZV are mixed and incubated for 1 h at 37 °C. (2) The mixture is added to cells in 6-well culture plates and incubated for 2 h at 37 °C, and then the culture medium is replaced. (3) After culturing for 5–7 days, the cells are fixed and stained to visualize the plaques, and the neutralization titer of serum samples can be calculated according to the number of plaques.

**Figure 4 microorganisms-11-00519-f004:**
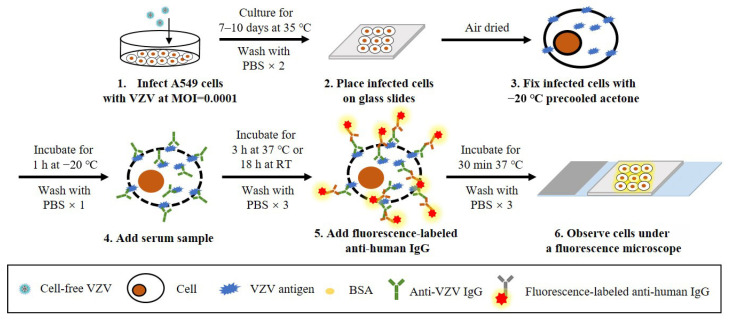
Schematic diagram of the protocol of a reported IFAT for the detection of anti-VZV antibodies. (1) Human lung carcinoma cells (A549) are infected with VZV at MOI = 0.0001 and cultured for 7–10 days at 35 °C. (2) After washing, the infected cells are placed on glass slides and air-dried. (3) Cells are fixed with precooled acetone for 1 h at −20 °C. (4) After washing, diluted serum samples are added and incubated for 3 h at 37 °C or for 18 h at room temperature (RT). (5) After washing, fluorescence-labeled anti-human IgG is added and incubated for 30 min at 37 °C. (6) After washing, the stained cells are observed under a fluorescence microscope.

**Figure 5 microorganisms-11-00519-f005:**
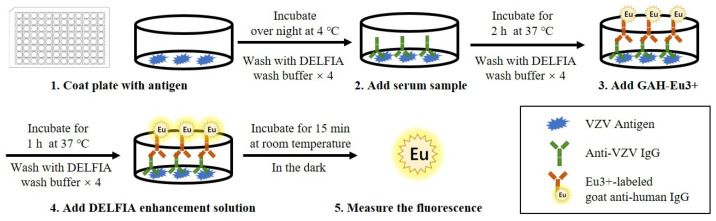
Schematic diagram depicting the basic steps of TRFIA for the detection of anti-VZV antibodies. (1) DELFIA microtiter plates are coated with VZV antigen and incubated overnight at 4 °C. (2) After washing, diluted serum samples are added to the plates and incubated for 2 h at 37 °C. (3) After washing, europium (Eu3+)-labeled goat anti-human IgG (GAH-Eu3+) is added and incubated for 1 h at 37 °C. (4) After washing, DELFIA enhancement solution is added and incubated for 15 min at room temperature in the dark. (5) The plate is read using a DELFIA 1234 reader to measure the fluorescence intensity of EU3+.

**Figure 6 microorganisms-11-00519-f006:**
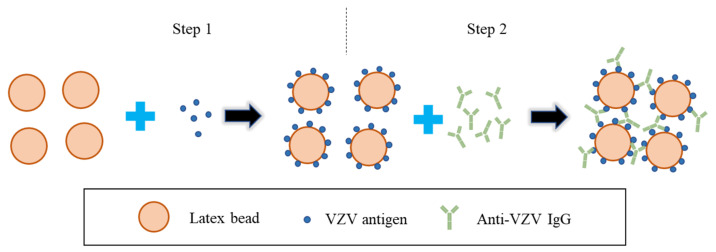
Schematic diagram depicting the basic steps of LA for the detection of anti-VZV antibodies. (1) The latex beads are coated with VZV antigen (e.g., purified VZV gE). (2) Serially diluted serum samples are added to the synthetic latex beads coated with VZV antigen, and the test samples are determined as positive by the observation of the agglutination reaction.

**Figure 7 microorganisms-11-00519-f007:**
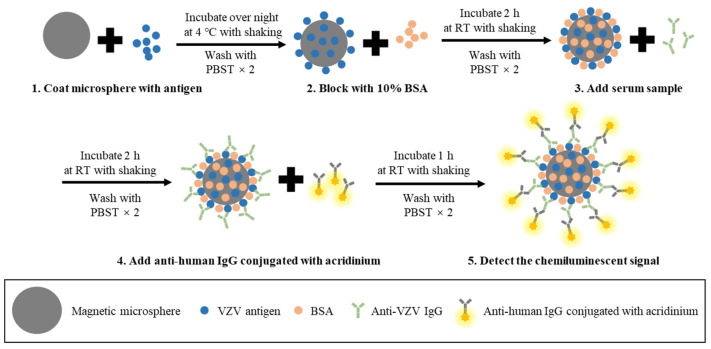
Schematic diagram depicting the basic steps of CLIA for the detection of anti-VZV antibodies. (1) Magnetic microspheres are coated with the VZV antigen (e.g., VZV gE) and incubated overnight at 4 °C with shaking. (2) After washing, microspheres are blocked with 10% BSA and incubated for 2 h at 37 °C with shaking. (3) After washing, diluted serum samples are added to the coated microspheres and incubated for 2 h at 37 °C with shaking. (4) After washing, anti-human IgG conjugated with acridinium is added and incubated for 1 h at room temperature (RT) with shaking. (5) After washing, the magnetic microspheres are resuspended with PBS and the chemiluminescent signal is detected with a chemical luminescence immune analyzer.

**Figure 8 microorganisms-11-00519-f008:**
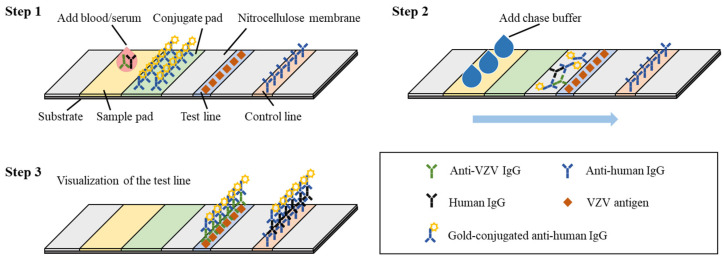
The fabricated LFIA strip for the detection of anti-VZV antibodies comprises of a base nitrocellulose membrane, a sample loading pad, a conjugate pad with immobilized gold-conjugated anti-human IgG, a test line with pre-absorbed VZV antigen (e.g., VZV gE) and a control line that serves to confirm if the strip is working. The basic steps of LFIA are as follows: (1) Add 15 to 20 μL of whole blood or serum to the sample pad of the strip. (2) Add three drops of chase buffer to allow migration of gold-antibody complexes onto the test line and the control line. (3) Anti-VZV antibodies in the sample are captured by the VZV antigen in the test line, and the presence of both a control line and a test line is used to define a VZV-positive test result.

**Table 1 microorganisms-11-00519-t001:** Comparison of four types of FAMA tests for detecting anti-VZV antibodies.

Type	Characteristics	Advantages	Limitations
Classic FAMA	Using live VZV-infected cells	High sensitivity and specificity. Gold standard for VZV antibody detection.	Tedious procedure. Subjective interpretation.
Fixed-cell FAMA	Using chemically fixed VZV-infected cells	High sensitivity. FAMA slides can be stored for a long time. Reduced hands-on time.	Tedious procedure. Subjective interpretation. Supposedly low specificity due to possible cross reaction with anti-HSV antibodies.
gE FAMA	Using live cells expressing VZV gE	High sensitivity and specificity. No contact with infectious VZV during operation.	Tedious procedure. Subjective interpretation. Lacking further validation. Unclear cutoff value.
Flow FAMA	Flow cytometry-adapted	Objective and automated measurement.	Special equipment is required. Unclear cutoff value.

**Table 2 microorganisms-11-00519-t002:** Comparison of ELISAs for detecting anti-VZV antibodies.

Type	Characteristics	Advantages	Limitations
WC-ELISA	Using whole lysates of VZV-infected cells as antigens	Commercially available. Antibodies against all VZV antigens can be detected.	Not sensitive enough to measure antibody responses to chickenpox vaccination.
gp-ELISA	Using purified VZV glycoproteins as antigens	Commercially available. Higher sensitivity and specificity than WC-ELISA.	Not sensitive enough to measure antibody responses to chickenpox vaccination (expect the Merck gp-ELISA). High cost of glycoprotein purification.
gE-ELISA	Using purified VZV gE as an antigen	Higher sensitivity and specificity than WC-ELISA.	Not commercially available. Only test for anti-gE antibodies.
Double antibody sandwich competitive ELISA	Capture antibody: anti-ORF9 antibody Detection antibody: HRP-labeled anti-gE antibody	Comparable sensitivity and specificity to FAMA	Not commercially available. Lacking further validation.
Double gE antigen sandwich ELISA	Using purified VZV gE as the coating antigen and HRP-labeled gE as the detection antigen.	Comparable sensitivity and specificity to FAMA	Not commercially available. Lacking further validation.

## Data Availability

Not applicable.

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
