# Peer review of "Current Methods for the Detection of Antibodies of Varicella-Zoster Virus: A Review"

_microorganisms, 2023, doi:10.3390/microorganisms11020519_

Round 1

Reviewer 1 Report

the work provides a complete picture of Current methods for the detection of antibodies to varicella-2 zoster virus in one place, which makes it much easier for medical practitioners to choose diagnostic methods. A collection of useful information collected in one place and well processed

Reviewer 2 Report

This manuscript presents a review of the different methods used to detect antibodies to varicella- 2 zoster virus. However, other methods that can be used for virus detection are worth mentioning. Although they are not methods to detect antibodies, they are also part of the information related to the subject, such as the Recombinase-Aided Amplification-Lateral Flow System. The authors could consider incorporating this additional information to enrich the document. If the subject is of interest, possibly information related to each technique could be presented with a graphic description that helps to understand the process better, especially for people new to the subject.

Reviewer 3 Report

This review discusses different methods for detection of antibodies that can be used to evaluate VZV infection rate among communities. The review provides a general overview of the most common (classical) methods for Ab detection and discuss their working principle, limitations, and application in detection of VZV antibodies. The review is well organised and suitable for publication after considering the following suggestions

1- The review would benefit from having a schematic depiction of the discussed techniques. These can be gathered in one scheme to demonstrate the work flow of the methods described in this review. 

2- It would be great to include a section for non-clssical techniques like  lateral flow assays or electrochemical assays. Some of these are available for antibodies and can be easily applied to VZV (for example 10.1016/j.jim.2023.113429 ). 

Reviewer 4 Report

- Authors in the Introduction should capture the burden of VZV more broadly to create a sense of an unmet need. A potential tangent could link the highly infectious and self-limiting nature of the disease along its complications with estimates of global burden and epidemiological susceptibilities (age groups affected, seasonality). This would augment the rationale of the study. An interesting outlook which authors should embark in order to make their study timelier and more topical, pertains to the advent of vaccination against COVID-19 in order to control the pandemic. Particularly, evidence suggests that VZV reactivation is associated with COVID-19 vaccination (doi.org/10.3390/vaccines9060572, doi.org/10.3390/vaccines9091013) with potentially higher susceptibility across older individuals (http://doi.org/10.1186/s12979-019-0164-9).

- Authors have attempted to outline in full different methods for the detection of VZV antibodies, however their description in terms of methodological and conceptual limitations of these techniques appears ill-defined. Examples of these are cases where authors become reserved to comments of “tedious procedure”, “subjective interpretation”, “lack of sensitivity”, “labour-intensive” and so on. The former is reflected in the future perspectives described within the Discussion whereby authors propose new developed techniques with very generic specifications such as “rapid assays with lower cost, better ease of use, and the necessary sensitivity, accuracy and reliability”. Although these simple descriptions are fine and do not impede the reader’s understanding, I believe that authors could augment further their discussion in terms of future perspectives by becoming more insightful, so that the derived conclusions as to what should be put forward become clearer.

Round 2

Reviewer 4 Report

The authors have now fulfilled most of my concerns.